# Vigilance in the Decision-Making Process Regarding Termination of Pregnancy Following Prenatal Diagnosis of Congenital Heart Disease—Application of the ‘Conflict Decision-Making Model’

**DOI:** 10.3390/ijerph19159137

**Published:** 2022-07-26

**Authors:** Yulia Gendler, Einat Birk, Nili Tabak, Silvia Koton

**Affiliations:** 1Department of Nursing, School of Health Sciences, Ariel University, Ariel 40700, Israel; 2Department of Nursing, School of Health Professions, Tel-Aviv University, Tel-Aviv 69978, Israel; tabak.nili@gmail.com (N.T.); koton@tauex.tau.ac.il (S.K.); 3Pediatric Heart Institute, Schneider Children’s Medical Center of Israel, Petah-Tikva 4920235, Israel; ebirk@clalit.org.il; 4Sackler Faculty of Medicine, Tel-Aviv University, Tel-Aviv 69978, Israel

**Keywords:** termination of pregnancy, congenital heart disease, decision-making process, conflict decision-making model, vigilance

## Abstract

The decision-making process regarding termination of pregnancy following prenatal diagnosis of congenital heart disease is a stressful experience for future parents. Janis and Mann’s conflict decision-making model describes seven ideal stages that comprise vigilant information-gathering as an expression of the qualitative decision-making process. In our study, we attempted to determine whether parents who face the decision regarding termination of pregnancy undertake a qualitative decision-making process. Data were collected over 2-year period using structural questionnaires. The sample consisted of two hundred forty participants; sixty-nine (28.75%) declared that their decision was to terminate the pregnancy. A significant difference in the quality of the decision-making score was noted between parents who decided to continue with the pregnancy vs. parents who opted for termination (mean score of 10.15 (5.6) vs. 18.51 (3.9), respectively, *p* < 0.001). Sixty-two (90%) participants within the termination of pregnancy group went through all seven stages of vigilant decision-making process and utilized additional sources for information and consultation. Parents who decided to continue with the pregnancy made swift decisions, often without considering the negative and positive outcomes; this decision-making pattern is considered non-vigilant and ineffective. Identification of future parents at risk of going through an ineffective decision-making process may help health professionals to determine the best way to provide them with information and support.

## 1. Introduction

Decision-making is a universal process. People in all cultures encounter recurring problems and opportunities that require significant choices between competing alternatives. Over the years, many theories have attempted to describe how people make decisions. The most familiar theoretical model for normative decision-making under uncertainty is the expected utility model [1]. This model assumes that people adopt rational behaviour, and make decisions according to expectations of utility. In other words, when faced with several actions, each of which could give rise to more than one possible outcome, all with different probabilities, the rational procedure is to identify all possible outcomes, determine their values (positive or negative), estimate the probabilities of different results from each course of action, and multiply the value of the outcome and its probability of occurrence to give an ‘expected value’, or the average expectation for an outcome.

Naturalistic decision-making theories [2] state that automatic or ‘gut reactions’ are important, prevalent, and may be efficient in the decision-making process. These theories focus on how people make tough decisions under difficult conditions, such as limited time, uncertainty, high stakes, vague goals, and unstable conditions. The theories conclude that during the decision-making process, people do not generate or compare option sets but instead use prior experience to rapidly recognize situations in the forms of repertoires of patterns. These patterns highlight the most relevant cues, provide expectancies, identify plausible goals, and suggest typical types of reactions in similar situations. When people need to make a decision, they can quickly match the situation to the patterns they have learned [3]. However, in most cases, people don’t have prior experience with decision-making regarding termination of pregnancy. Thus, naturalistic decision-making theories cannot be fully adopted in research on this topic.

Previous studies acknowledged that decisions following prenatal diagnosis of congenital heart disease are not necessarily rational [4,5]. Most parents do not weigh and quantify risk and burden in a mathematical and logical way. Rather, they seek ‘the least loss’ option that takes into account, for example, the perception of their ability to cope with a child with a heart defect [5]. Moreover, most of the decision-making theories do not recognise at least some of the factors that contribute to the decision to be made following the prenatal diagnosis. These factors include the short time available for decision-making, the fact that more than one individual would be involved in the decision, and the extremely high stakes associated with such a decision.

Nowadays, shared decision-making process is the most common pattern among future parents with fetuses diagnosed with congenital heart disease. However, the process itself may differ between the couples. Some couples make a swift decision whether to continue or terminate the pregnancy, but some need more time to process the received information and plan their next steps; these couples gather information from various sources, weigh the pros and cons of each alternative, perform additional tests (e.g., amniocentesis to rule out chromosomal abnormalities). Despite these differences, most couples reported that the decision-making process was the most challenging part and when they were finally able to make the decision, they felt relieved [6].

Janis and Mann (1977) attempt to provide a comprehensive descriptive theory of how people cope with vital decisions. Janis and Mann see the human as a ‘reluctant’ decisionmaker—beset by conflict, doubts, and worries. They present a ‘conflict model’ of decision-making, one applicable to all stressful, affect-laden situations in which ‘hot’ cognitions are at work. They suggest that decisions following seven ‘ideal’ procedural criteria are more likely to result in a decision that follows the objectives of the decisionmaker. The process followed by the decisionmaker, to the best of his or her ability and within his or her information-processing capabilities, includes the following steps, which would be completed in the described order:1.Thoroughly canvasses a wide range of alternative courses of action;2.Surveys the full range of objectives to be fulfilled and the values implicated by the choice;3.Carefully weighs whatever he knows about the costs and risks of negative and positive consequences which could flow from each alternative;4.Intensively searches for new information relevant to further evaluation of the alternatives;5.Correctly assimilates and takes account of any new information or expert judgment to which he or she is exposed, even when the information relevant to the judgment does not support the course of action that he or she initially prefers;6.Re-examines the positive and negative consequences of all known alternatives, including those originally regarded as unacceptable, before making a final choice;7.Makes detailed provisions for implementing or executing the chosen course of action, with special attention to contingency plans that might be required if various known risks were to materialize.

To meet these ideal criteria, the decisionmaker must have sought information about alternative courses of action, carefully weighed the risks, costs, and benefits of all the alternatives, and re-evaluated and explored the positive and negative consequences of these alternatives. A through information search and unbiased assimilation of new information which might lead to an effective and qualitative coping pattern is defined as vigilance. In state of extreme tension, other patterns of coping and decision-making come into effect, patterns which are less efficient and produce decision-making of poorer quality. These poorer decision-making patterns are: (i) unconflicted adherence—when the decisionmaker complacently decides to continue with his or her preliminary decision; (ii) unconflicted change—when the decisionmaker chooses the alternative that looks best, without examining other alternatives; (iii) defensive avoidance—when the decisionmaker makes a decision while selectively ignoring available information and devising rationalizations to refute information forced upon him or her; and (iv) hypervigilance—when the decisionmaker, threatened by imminent danger and with only a short decision time available, chooses what looks like the best alternative, without examining the consequences [7,8].

Few studies have, using Janis and Mann’s ‘conflict model’, analysed the decision-making process of continuation or termination of pregnancy following prenatal diagnosis of a fetal malformation. Swigar et al., (1977) described a situation in which pregnant women carrying fetuses with congenital defects preferred ‘not to know’ about a diagnosed defect. They refrained from thinking about the possible consequences of terminating a pregnancy, demonstrating poor-quality decision patterns, such as conflict-free persistence and conflict-free change [9]. Others adopted a defensive avoidance pattern, which leads women who consider abortion either to avoid professional consulting until late stages of the pregnancy, or to decide on abortion without consulting their partners or significant others [8].

Mann et al., (1998) suggested that the decision-making process is closely related to cultural and social factors. Learning from decision-making coping patterns of university students, they reported that students from Western countries (USA, Australia, and New Zealand) were more confident in their decision-making abilities, compared to students from East Asia (Japan, Hong Kong, and Taiwan). Furthermore, the East Asian students tended to adopt avoidant and hyper-vigilant styles of decision-making [10].

### Study Objectives

We attempted to examine whether couples who face the decision regarding continuation or termination of pregnancy undertake a vigilant information gathering process as an expression of a qualitative decision-making process as expressed by their adherence to the seven stages described in Janis and Mann′s conflict decision-making model (1977) and their utilization of different sources of information and consultation. Additionally, we sought to explore the influence of socio-demographic, cultural, and religious backgrounds on the decision-making process.

We hypothesized that:1.Parents who face the decision regarding continuation or termination of pregnancy following prenatal diagnosis of congenital heart disease will undertake a vigilant information gathering process, regardless of their decision.2.The decision-making process will be influenced by cultural and religious context; parents from the Jewish population and parents who define themselves as secular will undertake a vigilant information gathering process, parents from the Jewish population who define themselves as traditional or religious, and parents from the non-Jewish population will undertake a non-vigilant information gathering process.

## 2. Materials and Methods

### 2.1. Study Design

A study was conducted in the Institute of Pediatric Cardiology in the Schneider Children′s Medical Center of Israel. Pregnant women are referred to this center in order to perform fetal echocardiogram tests, either in the presence of maternal risk factors (i.e., family history of congenital heart disease, gestational diabetes, infections acquired in pregnancy) or when routine obstetric ultrasonography suggests a potential risk of congenital heart defect in the fetus (i.e., abnormal nuchal translucency screening, or after an early anatomy scan). These reasons for referral are consistent with international standards on indications for fetal echocardiogram based on the usual risk factors [11]. Purposive sampling was used for recruitment; medical records of the Institute of Pediatric Cardiology at Schneider Children′s Medical Center of Israel were reviewed and lists of potential participants were produced in advance. Using these lists, we were able to effectively select the appropriate sample and avoid selection bias.

### 2.2. Setting and Procedures

The participants were recruited over a two-year period—between December 2014 and December 2016. Fetal echocardiograms were carried out by a senior pediatric cardiologist at the Institute of Pediatric Cardiology. As a standard of care, and according to the findings, information on a broad variety of topics, including the diagnosis with associated consequences, natural history of the disease, possible surgical treatment, and prognosis was provided by the pediatric cardiologist. In cases of diagnosis of a fetal cardiac malformation, the parents were offered the opportunity to participate in the present study, regardless of their decision to continue the pregnancy or terminate it. The potential participants were given written and oral information about the study. Couples who agreed to participate completed a written informed consent and were then contacted by the first author (Y.G.) and asked to choose a suitable location to be interviewed for further data collection.

The data collection was carried out using structural questionnaires. The questionnaires were filled out individually by each participant 2–4 weeks after the fetal echocardiography exam that had revealed a congenital heart disease. In order to better understand the decision-making process, the participants were asked to fill out the questionnaire without consulting with his/her partner. The study was preceded by a pilot study on 26 participants, in order to validate the research questionnaires.

### 2.3. Participants

The sample consisted of 120 couples (240 participants), each with a fetus that had been diagnosed with a congenital heart defect, defined as a condition in need of surgical treatment, at the Institute of Pediatric Cardiology in Schneider Children′s Medical Center. Couples who were referred by a medical tourism agent, or who did not speak Hebrew, Arabic, or Russian were excluded from the study. Other exclusion criteria included diagnoses of chromosomal abnormalities or extra-cardiac malformations.

### 2.4. Definition of Variables

Socio-demographic information from the participants (i.e., age [years], origin, ethnicity, religious affiliation, nationality, and highest education achieved) was collected using the study questionnaires and the medical records. The independent variable, the decision regarding continuation or termination of pregnancy, was measured by two direct questions: ‘What was your shared decision?’ and ‘What was your personal decision?’ Couples’ shared decisions did not necessarily reflect each person’s personal decisions; thus, the personal decision of each participant was taken into account in data analysis. Two groups of participants were created according to their termination decisions: the ‘continuation of pregnancy’ group, and the ‘termination of pregnancy’ group.

The dependent variable, the vigilance of the decision-making process, was measured using a structural validated questionnaire; the questionnaire was based on the Decision Making Quality Scale (DMQS), which assesses the degree to which a person adheres to seven qualitative criteria of decision-making as determined by Janis and Mann (1977) [8]. The questionnaire asked individual decisionmakers to evaluate their performance at each stage of the decision-making process, ranking their performance from 0 (‘Not at all true’) to 3 (‘Very true’) according to the level of their agreement/disagreement with a series of statements. The Hebrew version of the questionnaire was validated and used in a study by Rabin and Tabak (2006). The reported internal reliability of the Hebrew version was 81 on a Cronbach′s Alpha test [12]. We used the validated Hebrew version and assessed its internal reliability in our study population (α = 0.93). The final scores were obtained by using the data from questionnaire, multiplying the number of stages (seven) by the score for each stage (0–3). The range thus ran from a low 0 to a high of 21 and were defined as ‘total adherence index’. The scores were grouped into two categories: a score of from 0 to 14, which signified a ‘non-quality decision,’ and from 15 to 21—a ‘quality decision’ [8,12].

### 2.5. Potential Bias

Several measures were taken in order to minimize potential selection and information biases; active selection of the participants from the medical records of the Institute of Cardiology enabled us to avoid selection bias. Another means of reducing the possibility for selection bias was the exclusion of parents whose fetus was diagnosed with chromosomal or extra-cardiac abnormality in addition to congenital heart disease. The study design minimized the possibility for recall bias. Data collection was performed in proximity to the enrollment (2–4 weeks), in order to minimize possible information bias.

### 2.6. Statistical Methods

Two groups were created according to the individual decisions of the participants: the ‘continuation of pregnancy’ group and the ‘termination of pregnancy’ group. The distribution of variables by decision group was presented as means (SDs) for continuous variables, and as frequencies (%) for categorical variables. The difference between the two groups in the quality of their decision-making scores was analyzed using an independent samples *t*-test. Differences in the utilization of sources of information and consultation between the two groups were analyzed using a χ^2^ test. Subsequently, we conducted logistic regression analysis to compute the odds ratios and 95% confidence intervals (CIs) for vigilant decision-making process with gender, religious affiliation, population group, years of education, and decision regarding TOP as predictor variables. Analyses were performed using the IBM SPSS software (Version 24, Armonk, NY, USA). All tests were two-tailed and *p* values < 0.05 were considered to be statistically significant.

## 3. Results

Two hundred couples met the inclusion criteria and were offered to participate in the study. Thirty-two couples refused to fill the questionnaires, twenty-five couples were excluded from the study during the data collection process (twenty-three excluded due to prenatal diagnosis of chromosomal abnormalities and/or extra-cardiac malformations in the fetus, and two excluded due to language barriers). Twenty-three couples signed the informed consent form and weren’t available to be interviewed during the 2–4 weeks period defined in the study protocol. The final sample consisted of one hundred twenty couples (two hundred forty participants, 88.8% Jews, 9.2% Muslim-Arabs and 2% Christians, Bedouins, or Druze). Seventy-two (30%) participants defined themselves as secular, seventy-four (30.8%) as traditional and seventy-four (30.8%) as religious, and twenty (8.4%) participants were Jewish-Orthodox. The distribution by population group and religious affiliation in our sample reflects the distribution of the population in Israel. Socio-demographic characteristics of the two hundred forty participants are presented in Table 1.

Following prenatal diagnosis of congenital heart disease, sixty-nine (28.8%) participants declared that their decision was to terminate the pregnancy, and one hundred seventy-one (71.2%) decided to continue with the pregnancy. Forty-seven (27.5%) participants within the ‘continuation of pregnancy’ group made an immediate decision on the day of diagnosis, and one hundred forty-eight (86.6%) out of 171 participants declared that the decision was made within a week or less. Within ‘termination of pregnancy’ group, on the other hand, only ten (6.9%) out of sixty-nine made an immediate decision and fifty-four (78.3%) participants made the decision within 1–2 weeks.

Differences in the total score of vigilant information processing between the ‘continuation of pregnancy’ group and the ‘termination of pregnancy’ group are shown in Figure 1.

The box plot shows that among the ‘continuation of pregnancy’ group, there were subjects with a decision-making score of zero, meaning that they did not go through any of the seven stages that comprise a vigilant decision-making process. The median score of the quality of the decision-making process in the ‘continuation of pregnancy’ group was 11, and the third quartile was 14. In other words, the score for the decision-making processes of 75% of the participants in this study group was low. Within the ‘termination of pregnancy’ group, there were no subjects whose decision-making process score was zero. The lowest score of a single participant was three. Most scores in this group were concentrated at the maximum limit, with a median of 21, the highest score that could be obtained. Apart from seven individual subjects, the minimum score in this group was 15, which, according to Hollen (1994) and Rabin and Tabak (2006), indicates a qualitative decision-making process.

Figure 2 and Figure 3 demonstrate the differences between the ‘continuation of pregnancy’ group and the ‘termination of pregnancy’ group in their utilization of sources of information and consultation.

Forty-one percent of the ‘termination of pregnancy’ group stated that they used more than one source of information, such as informative websites, professional literature, and social media, prior to making the decision (vs. 15% within the ‘continuation of pregnancy’ group). Fifty-nine percent of the participants in the ‘continuation of pregnancy’ group didn’t look for additional information regarding the heart disease of the fetus (vs. 25% within the ‘termination of pregnancy’ group).

Forty percent of the participants within the ‘continuation of pregnancy’ group stated that they didn’t consult with another person prior to making the decision (vs. 13% within the ‘termination of pregnancy group’). The ‘continuation of pregnancy group’ had a greater proportion of participants who consulted with extended family members (23% vs. 7% within the ‘termination of pregnancy’ group) or with a spiritual advisor (12% vs. only 2% within the ‘termination of pregnancy’ group).

The logistic regression model delineated four major factors (gender, religious affiliation, years of education, and TOP decision) that are associated with a vigilant decision-making process. Vigilant decision-making processes were 6.49 more likely (95% CI [2.85, 14.78]) for female participants, and 2.51 more likely (95% CI [1.14, 5.53]) for secular participants as compared for participants with other religious affiliations. In addition, one year of education was associated with vigilant decision-making processes (OR-1.31, 95% CI [1.17, 1.71]), and participants who decided on TOP were 64.23 more likely (95% CI [19.83, 99.71]) to undertake a vigilant decision-making process (Table 2).

## 4. Discussion

In this research, the decision-making process of prospective parents regarding termination of pregnancy following prenatal diagnosis of congenital heart disease was analysed using Janis and Mann′s (1997) conflict decision-making model. This theoretical model delineates a rational or quality decision-making style known as vigilance, which is characterized by systematic search for information, careful consideration of all viable alternatives, and unhurried, non-impulsive making of the final decision [13]. The key findings of our study were that female gender, secular religious affiliation, and higher education were significantly associated with a vigilant decision-making process regarding termination of pregnancy. Furthermore, couples who decided on termination of pregnancy following prenatal diagnosis of congenital heart disease underwent a vigilant decision-making process, as compared to couples who decided to continue with the pregnancy; parents who decided to continue with the pregnancy made swift decisions, always within a week or less. Parents who opted for termination of pregnancy underwent more in-depth information gathering processes; they utilized more sources of information (e.g., social media, professional literature, and websites) and consulted with other families that had a child with heart disease, and with medical professionals.

The unexpected nature of a prenatal diagnosis of congenital heart disease means that decision-making about whether or not to continue a pregnancy usually takes place at a time of crisis and distress [14]. According to Janis and Mann (1977), the need to make a decision inherently involves a conflict which engenders a certain degree of stress, the excess or absence of which is in turn a major determinant of the subject′s failure to make a good decision, since it is associated with unproductive information search, assessment and decision-making patterns. The results of our study revealed that parents who decided to terminate a pregnancy following a prenatal diagnosis of congenital heart disease adhered to all seven stages that comprise a qualitative decision-making process. They actively searched for additional information about the diagnosis, consulted with several professional and social sources (i.e., social media), and carefully considered the positive and negative consequences of each alternative prior making the decision. Parents who decided to continue with the pregnancy, on the other hand, adopted inferior decision-making patterns. They were generally content with the information provided by their cardiologists, and most of them preferred not to look for additional information and not to seek advice. Couples who chose to share their deliberations with others turned to their extended families or spiritual advisors (i.e., Rabbi, Pastor, or Imam). Rabin and Tabak (2006) found in their study that people suffering from life-threatening conditions generally failed to carry through an adequate decision-making process. The researchers surmised that decisions that are made in ‘emergency’ conditions caused physical and mental blocks which impaired the ability of the patients to conduct a vigilant information gathering; instead, they passed the responsibility for their fate onto their family or their physician. The same pattern may be present here; the ‘emergency’ of the decision regarding the fate of the pregnancy intervenes in the decision-making process and forces parents to deliberately ignore information, to devise rationalizations to refute information provided by the physicians, and to let others decide for them. This type of behavior is what Janis and Mann (1977) termed ‘defensive avoidance’.

In many societies, decisions on the termination of pregnancy are influenced by social, cultural, and religious factors [15,16]. Furthermore, the entire process of decision-making is associated with social and cultural contexts as well [17,18]. These findings are reflected in our results, with substantial differences in the characteristics of the study groups; the ‘continuation of pregnancy’ group was mostly comprised of participants who defined themselves as traditional or religious, with primary or secondary education, while most of the participants within the ‘termination of pregnancy’ group were secular Jews with an academic education. These characteristics directly influenced the decision-making pattern; parents who decided to continue with the pregnancy underwent non-vigilant and non-qualitative decision-making processes. Another possible explanation for this finding is the fact that many traditional societies (Muslim Arabs and Jewish Orthodox) decline the option of termination of pregnancy [19,20,21]. This attitude towards termination makes the vigilant decision-making process superfluous.

A surprising finding in our study was the low application rate for consultation with other families or a second medical opinion by parents who decided to continue with the pregnancy. Previous studies reported that parents’ support groups, blogs, and websites are important sources of support for parents after receiving the diagnosis of congenital heart disease in a fetus. Medical professionals (physicians and nurses) were described as the main sources for information and reassurance [22,23,24]. Our findings can be explained by the limited time frame for recruiting participants for the study (2–4 weeks after the initial diagnosis). Parents who decided to continue with the pregnancy may seek support and consultation on later stages of the pregnancy.

### 4.1. Strengths and Limitations of the Study

The design of the study ensured adequate and comprehensive data collection. We interviewed parents shortly after the decision regarding termination or continuation of the pregnancy, minimizing the possibility of recall bias. The data collection was carried out for a two-year period, enabling medical and demographical diversity of the participants.

However, our study has limitations. First, it was a single center study conducted at the Institute of Pediatric Cardiology at Schneider Children′s Medical Center of Israel. Even though our sample included couples with diverse cultural backgrounds, two couples were excluded since they could not be interviewed in Hebrew, Arabic, or Russian.

The proximity of the data collection to the decision regarding continuation or termination of pregnancy precludes study of the long-term outcomes of parental decisions (such as regret, struggling, and management style). This may serve as a topic for future research.

Accurate understanding of the diagnosed condition is an important component of decision-making following prenatal diagnosis of congenital heart disease [14]. However, people differ in the amount of information they want or need in order to make a decision [25]. The model proposed by Janis and Mann (1977) does not take this into account and unequivocally determines that individuals who did not search for additional information diminished the quality of their decision-making processes.

### 4.2. Implications for Practice

Parents receiving a prenatal diagnosis and facing difficult decisions differ in the amount of information they have, as well as the support they can receive to address their needs. Parents who opted for termination of pregnancy carefully considered other alternatives and, unhurried, non-impulsively, made their final decision. Parents who decided to continue with the pregnancy, on the other hand, made swift decisions, without considering the negative and positive outcomes of all the alternatives; this decision-making pattern is, according to Janis and Mann (1977), non-vigilant and non-qualitative. The decision-making processes were influenced by cultural and social factors. These findings have implications for the counseling process following a prenatal diagnosis of congenital heart disease. The health professionals involved must identify those at high risk of non-quality decision-making processes and determine the best way to provide them with information and support. Parents should be given information on pre-approved websites, connected with support groups, and given the opportunity to be placed in contact with other parents who have a child with a similar diagnosis. In this manner, the health professionals can lead the parents to more vigilant and qualitative decision-making processes.

## 5. Conclusions

Our study analyzed the parental decision-making process following a prenatal diagnosis of congenital heart disease, using a decision-making model proposed by Janis and Mann (1977). This is a descriptive model of the internal conflict involved in the individual decision-making process, and the decision patterns assessed by the seven stages of vigilance correspond to possible courses of action that a subject may follow in response to this internal conflict. Integration of this model in parental decision-making process regarding termination of pregnancy following a prenatal diagnosis of congenital heart disease highlights the differences in quality of the decision-making process between parents who decide on termination vs. parents who decide on continuation of pregnancy, taking into account cultural and social contexts.

## Figures and Tables

**Figure 1 ijerph-19-09137-f001:**
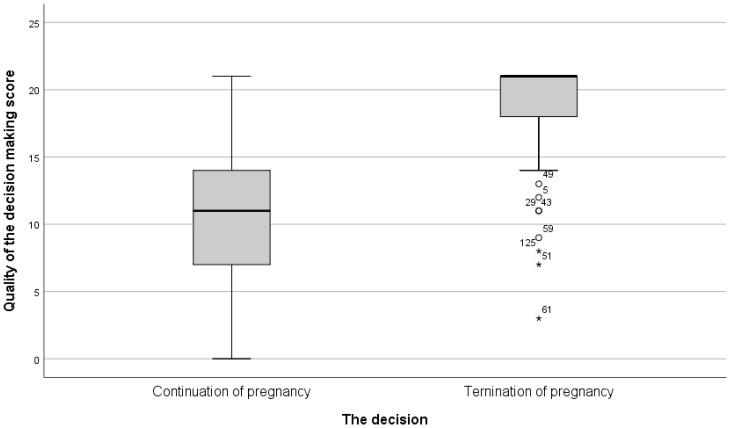
Differences in the total scores of ‘vigilant information processing’ between ‘continuation of pregnancy’ and ‘termination of pregnancy’ groups. *t* = 13.587, *p* < 0.001. * Mean and median scores of the quality of the decision-making process: ‘Continuation of pregnancy’ group: M (SD) = 10.15 (5.6); Med [range] = 11 [0–21], ‘Termination of pregnancy’ group: M (SD) = 18.51 (3.9); Med [range] = 21 [3–21].

**Figure 2 ijerph-19-09137-f002:**
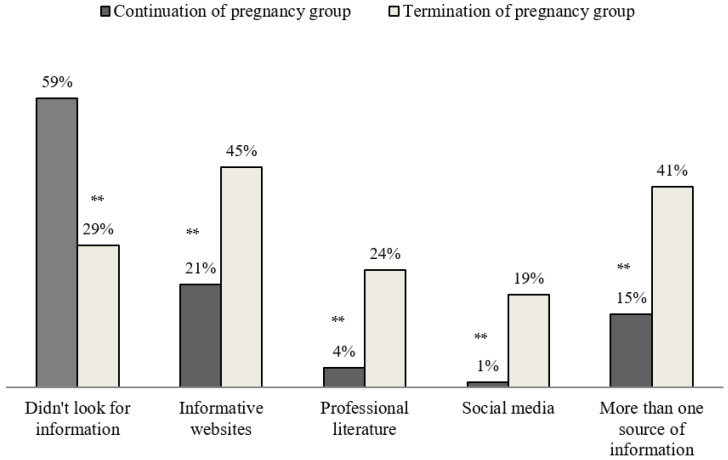
Utilization of information sources—comparison between ‘continuation of pregnancy’ and ‘termination of pregnancy’ groups; ** *p* < 0.001.

**Figure 3 ijerph-19-09137-f003:**
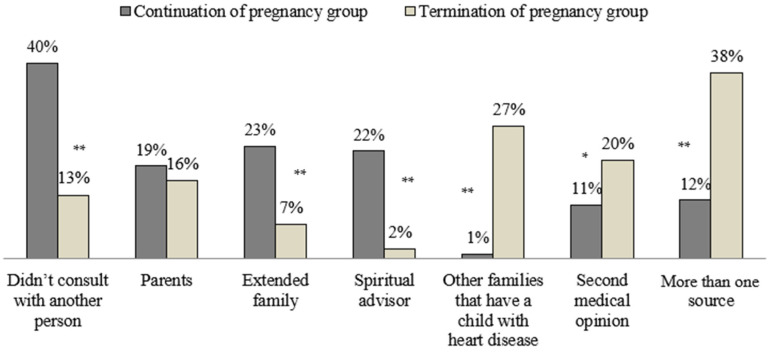
Utilization of sources for consultation—comparison between ‘continuation of pregnancy’ and ‘termination of pregnancy’ groups; * *p* < 0.05, ** *p* < 0.001.

**Table 1 ijerph-19-09137-t001:** Socio-demographic characteristics of the study population (*n* = 240).

	*n* (%) or Mean (SD)
Gender	
Male	120 (50%)
Female	120 (50%)
Age	31.3 (4.5)
Population group	
Jews	213 (88.8%)
Muslim Arabs	22 (9.2%)
Other (Christian, Bedouin, Druze)	5 (2%)
Religious affiliation	
Secular	72 (30%)
Traditional	74 (30.8%)
Religious	74 (30.8%)
Jewish-Orthodox	20 (8.4%)
Higher education achieved	
Primary education	20 (8.4%)
Secondary education	61 (25.4%)
Diploma	56 (23.3%)
Academic degree	103 (42.9%)
Years of education	14.1 (2.3)

**Table 2 ijerph-19-09137-t002:** Logistic regression model of factors that influence vigilance in the decision-making process.

Factors	OR	95% CI		*p*
		Lower	Upper	
Gender				
Male	1 (Ref)			
Female	6.49	2.85	14.78	<0.001
Population group				
Other	1 (Ref)			
Jews	1.13	0.31	4.17	0.86
Religious affiliation				
Other	1 (Ref)			
Secular	2.51	1.14	5.53	<0.001
Years of education				
Estimate for each year	1.41	1.17	1.71	<0.001
Decision to terminate the pregnancy				
No	1 (Ref)			
Yes	64.23	19.83	99.71	<0.001

CI—Confidence Interval; Ref—Reference. The scores of the decision-making quality scale were grouped into two categories: a score of from 0 to 14, which signified a ‘non-quality decision,’ and from 15 to 21—a ‘quality decision’.

## Data Availability

The data that support the findings of this study are available on request from the corresponding author. The data are not publicly available due to privacy concerns and ethical restrictions.

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
