# Peer review of "Vigilance in the Decision-Making Process Regarding Termination of Pregnancy Following Prenatal Diagnosis of Congenital Heart Disease—Application of the ‘Conflict Decision-Making Model’"

_ijerph, 2022, doi:10.3390/ijerph19159137_

Round 1
Reviewer 1 Report
The authors addressed a very interesting topic regarding decision making in case of cardiac anomaly.
The results are not surprise ling since low educational and more religious people tend to make swift, unsupported by documentation decisions.
The groups are homogeneous concerning cultural and religious background and I would appreciate if the authors can comment on how the questionnaire can be extended in general more heterogenic population and how can it improve the counselling process of couples affected by such condition.
Also I would recommend following those decision making couples in time to evaluate how content they are with the decision
Author Response
Thank you for your time and efforts in reviewing our manuscript. We found the comments and suggestions offered by you to be highly useful. Accordingly, we incorporated the appropriate changes within the current version of the manuscript. Point-by-point responses to each of the comments are provided below.
The distribution by population group and religious affiliation in our sample represents the distribution of the population in Israel, thus we believe that our findings could be generalized to the Israeli population (https://www.cbs.gov.il/en/subjects/Pages/Demographic-Characteristics.aspx) . This information was added in the first paragraph of the results section.
Thank you for the second comment, we agree that this is an important topic for further research. However, this study was not designed for follow up and accordingly the participants did not sign the informed consent for follow up.
Reviewer 2 Report
Dear authors,
it is an interesting piece of research work. I read it and found interesting perspectives of the topic.
A few points to address:
1. It is written that table 1 presents results regarding the nationality? I could not find it. Can you please assist?
2. The sample consists of almost 89% jewish polulation. Since religion is examined do you think that the sample is not equally distributed thus raising concerns of regarding statistical bias and results validity? Can you elaborate on that?
Author Response
Thank you for your time and efforts in reviewing our manuscript. We found the comments and suggestions offered by you to be highly useful. Accordingly, we incorporated the appropriate changes within the current version of the manuscript. Point-by-point responses to each of the comments are provided below.
- 'Nationality' refers to 'population group'. The terminology was modified in the text to maintain consistency.
- The distribution by population group and religious affiliation in our sample represents the distribution of the population in Israel, thus we believe that our findings could be generalized to the Israeli population (https://www.cbs.gov.il/en/subjects/Pages/Demographic-Characteristics.aspx).
This information was added in the first paragraph of the results section.
Reviewer 3 Report
Manuscript deals with a relevant topic with potential consequences for clinical work. Nevertheless, some concerns should be recognized.
Introduction section is a little confusing. authors often switch between cognitive process in decision making and transition to parenthood in case of prenatal diagnosis. It could make hard to understand how is the global back ground theory and how to it could be applied at the topic that was chosen
Methods. I think that the main weakness of the study is the absence of a standardized instrument for the assessment. The main aim of the study is assessed by a categorical variables, while the process of decision making by an instrument without name that we only know that was used in a previous study. It make very hard to understand results and their generalization. Many info are needed and it should be clearly declared in limit section.
Stastistical anlyses is very simple. Given the sample size, why did authors not use logistic regression or a model to understand the topic instead of X2?
Also in case of t test, I am not sure that variables were independent each other given they are part of a same processes. if authors verified assumption should declare it, or use a model that could better verify the interdependence of each item.
Quality of Results is influenced by the limit of statistical analyses. Furthermore, I did not understand why authors did not present result in the same order of the declaration of aims (They are inverted).
I also did not understand Table 1: why did authors compared sociodemographic variables? did it regards descriptive analyses? it should be declared. Interestingly, differences emerged in level of education: how authors consider this variable? is it controlled in subsequent analyses?
It should be note that Table 1 also included cultural variables that should be investigated in aim 2. So, if Table shows inferential analyses, all variables should be included as objective of the study, or descriptive and inferential analyses should be differentiated.
Finally, text in lines 303-320 is very argumentative for result section, and it is more appropriate for discussion section.
Please check also check guidelines for citation and references section
Author Response
Thank you for your time and efforts in reviewing our manuscript. We found the comments and suggestions offered by you to be highly useful. Accordingly, we incorporated the appropriate changes within the current version of the manuscript. Point-by-point responses to each of the comments are provided below.
Comment:
Introduction section is a little confusing. authors often switch between cognitive process in decision making and transition to parenthood in case of prenatal diagnosis. It could make hard to understand how is the global background theory and how to it could be applied at the topic that was chosen
Answer:
The introductory chapter was modified to allow more fluent reading and a better understanding of the existing theories compared to the theory we used in our research.
Comment:
Methods. I think that the main weakness of the study is the absence of a standardized instrument for the assessment. The main aim of the study is assessed by a categorical variables, while the process of decision making by an instrument without name that we only know that was used in a previous study. It make very hard to understand results and their generalization. Many info are needed and it should be clearly declared in limit section.
Answer:
The main outcome measure, vigilance of the decision-making process, was measured using a structural validated questionnaire that was based on the Decision-Making Quality Scale (DMQS). The scale assesses the degree to which a person adheres to seven quality criteria of decision making as determined by Janis and Mann and was validated in a large study of 766 participants by Hollen (Hollen PJ. Psychometric properties of two instruments to measure quality decision making. Res Nurs Health. 1994;17(2):137–48). This scale served as a basis for developing new scales and theories in shared decision making (Dy SM. Instruments for Evaluating Shared Medical Decision Making: A Structured Literature Review. Medical Care Research and Review. 2007;64(6):623-649. doi:10.1177/1077558707305941). Moreover, the Hebrew version of the scale was validated and used in a study on healthy participants (Rabin C, Tabak N. Healthy participants in phase I clinical trials: The quality of their decision to take part. J Clin Nurs. 2006;15(8):971–9). We apologize if this information was not clearly stated, the manuscript was modified for clarity.
Comment:
Statistical analyses is very simple. Given the sample size, why did authors not use logistic regression or a model to understand the topic instead of X2?
Answer:
Statistical analysis: Thank you for that comment, subsequently we added a logistic regression of factors that influence vigilance in the decision-making process.
Comment:
Also in case of t test, I am not sure that variables were independent each other given they are part of a same processes. if authors verified assumption should declare it, or use a model that could better verify the interdependence of each item.
Answer:
The table with the separate statements was removed.
Comment:
Quality of Results is influenced by the limit of statistical analyses. Furthermore, I did not understand why authors did not present result in the same order of the declaration of aims (They are inverted).
Answer:
A logistic regression model was added. The results are now presented in the same order as the aims.
Comment:
I also did not understand Table 1: why did authors compared sociodemographic variables? did it regards descriptive analyses? it should be declared. Interestingly, differences emerged in level of education: how authors consider this variable? is it controlled in subsequent analyses?
It should be note that Table 1 also included cultural variables that should be investigated in aim 2. So, if Table shows inferential analyses, all variables should be included as objective of the study, or descriptive and inferential analyses should be differentiated.
Answer:
We appreciate this comment, and we completely agree with the reviewer. Table 1 was modified and now presents the socio-demographic background of the participants. The Results section is now arranged in a logical order - starting with descriptive statistics of the sample characteristics and continuing with inferior statistics according to the research objectives.
Comment:
Finally, text in lines 303-320 is very argumentative for result section, and it is more appropriate for discussion section.
Answer:
Text in lines 303-320 was modified to suit the 'Results' chapter. The discussion on these findings are presented now only in the Discussion chapter.
Comment:
Please check also check guidelines for citation and references section
Answer:
Citations and references appear according to journal guidelines.
Round 2
Reviewer 3 Report
I thank authors for their work that significantly improved the manuscript. I had some further little considerations.
I had again some difficulties in correspondence between aims and results. If my understand is ok, t-test is used to assess aim 1, and logistic regression for aim 2. So, last part of results (figure 2 and 3) are part of which aim? the first one? if yes, please reorder results section. I think that aim 1 could be more describe, to respect all analyses that authors run.
I did not understand why authors included gender and level education in logistic regression analyses because they were not cited in aim 2. Please add them in aim, or explain why they were included in statistical analyses section.
Author Response
Thank you for your time and efforts in reviewing our manuscript. We found the comments and suggestions offered by you to be highly useful. Accordingly, we incorporated the appropriate changes within the current version of the manuscript. Point-by-point responses to each of the comments are provided below.
Figures 2 and 3 are part of the first aim, we reordered the results chapter according to the reviewers' suggestion and the figures now appear before the logistic regression. Additionally, we elaborated the first aim according to the reviewers' suggestion.
The socio-demographic characteristics were added to the second aim.